# Individual Differences in Parental Support for Numeracy and Literacy in Early Childhood

**Leanne Elliott \***, **Peter Zheng and Melissa Libertus**

Learning Research and Development Center, University of Pittsburgh, Pittsburgh, PA 15260, USA;
pez27@pitt.edu (P.Z.); libertus@pitt.edu (M.L.)
*   Correspondence: lek79@pitt.edu

**Abstract:** Past research has examined parental support for children's math and reading skills in the early years through parents' reports of their activities with their children in somewhat inconsistent ways. In this study, we use data from a large sample of parents (*n* = 259; 103 males) collected through Amazon's Mechanical Turk to examine dimensions of parental enrichment in both support for literacy and numeracy skills at home. Additionally, we examine how socioeconomic resources as well as parental beliefs relate to these dimensions of the home literacy and home numeracy environment. Factor analyses revealed two dimensions of literacy activities (i.e., passive and active literacy activities) and three dimensions of numeracy activities (i.e., numeracy applications, basic numeracy, and written numeracy activities). Income was positively associated with active literacy activities, whereas parents' educational attainment was negatively associated with active literacy activities and written numeracy activities. Additionally, parental beliefs, including their beliefs about the importance of literacy and math skills as well as their perceived responsibility for teaching their children reading, math, and language skills, related to home literacy and numeracy activities in distinctive ways. These results suggest that future research should explore parental enrichment practices with greater nuance, particularly when examining associations with socioeconomic status.

**Keywords:** home literacy; home numeracy; socioeconomic status; parent beliefs; individual differences



## 1. Individual Differences in Parental Support for Numeracy and Literacy in Early Childhood

Children's educational experiences at home, including activities involving reading or numeracy concepts, appear to foster early academic skill development. Practices to support children's language and literacy development, such as shared book reading [1], are often referred to as the home literacy environment (HLE) and predict a wide array of language and emergent literacy skills, such as vocabulary, phonological sensitivity, and word decoding [2–7]. Similarly, a growing number of studies have examined activities in the home that may promote children's numeracy development, or the home numeracy environment (HNE). These practices include a broad array of activities, such as counting, playing board games, and talking about money [8]. Despite the relative infrequency of these activities in the home compared to literacy activities [8–10], the home numeracy environment typically predicts children's math skills, although the existing literature is somewhat inconsistent compared to research on home enrichment more generally [11].

To complement the work examining links between the HLE and children's reading and math skills, a growing body of research has examined the factors that predict parents' provision of home literacy or numeracy activities. Past empirical work suggests that parents' socioeconomic status as well as their beliefs about math and reading are related to parents' home enrichment activities [12–15]. However, this body of literature has several notable limitations, as few studies account for the multifaceted nature of both the HLE and HNE or attempt to disentangle the effects of parents' educational attainment and household income as unique aspects of SES. In this study, we utilize a large sample of parents of

three- to six-year-old children to identify sub-dimensions of the home literacy and home numeracy environment. Additionally, we examine how factors such as parental income, educational attainment, and beliefs about math and reading relate to these dimensions.

## 2. Theoretical Dimensions of Home Enrichment

Sénéchal and LeFevre's Home Literacy Model [16] distinguished between two types of practices in the home to support language and literacy development: informal and formal literacy activities. These two domains of the HLE, in turn, predict reading achievement through distinct mechanisms. Informal literacy activities, such as shared book reading, were shown to predict children's language skills, including vocabulary and comprehension, whereas parents' formal, didactic activities fostered emergent literacy skills such as letter recognition. This model was among the first to unpack the complexity and nuances of parents' practices to support reading skills and has since been well validated by subsequent research [5,6,16,17].

The work examining formal and informal HLE activities inspired a complementary model of parental influences on numeracy achievement. In the Home Numeracy Model, Skwarchuk, Sowinski, and LeFevre [18] argue that two similar formal and informal dimensions characterize activities to support children's math learning. They found that formal math activities, or those with the explicit purpose of teaching children math concepts, fostered children's formal math skills (e.g., counting, numeral identification), whereas informal activities, or those that include math content incidentally, promoted informal or non-symbolic math skills, such as non-symbolic addition and subtraction skills. These dimensions of formal and informal math activities are frequently measured in past work [8,19,20], yet several methodological concerns regarding these measures have been raised [11]. For example, the pattern of associations between formal and informal aspects of the HNE and children's math achievement described is not consistently documented [8,19]. This may be due to differences in how the activities are measured across studies. Although other dimensions of the HNE have been used in past work, such as distinguishing between basic math concepts (e.g., counting) and advanced or complex math content (e.g., ordinal relations between numbers) [21–23], most work has focused primarily on formal and informal domains of the HNE.

## 3. SES and Home Enrichment

Differences in the home enrichment experiences of children from low- and high-SES backgrounds during the early years have been well documented in past research [12,14]. Composite measures of SES positively predict parental language input, including the overall amount of language input as well as grammatical complexity, which may support children's language learning [24–26]. Income also positively relates to the number of books in the home and frequency of literacy activities (e.g., shared book reading), such that these activities occur less often in low-income homes compared to high-income homes [27,28]. One study even found that these family-level literacy processes outweighed school and neighborhood characteristics in explaining SES disparities in reading skills throughout early elementary school [29].

Although fewer studies have examined SES differences in the HNE, some work suggests that parents from low-SES backgrounds engage in activities to support children's numeracy skills less frequently than their peers from high-SES backgrounds. One early study found that middle-class families engaged in qualitatively different types of activities than their working-class peers, including engaging in a broader range of activities and activities with more complex goals [30]. More recently, DeFlorio and Beliakoff [31] also found small but significant income differences in the range of parent-reported math activities in the home, such that parents with higher income engaged in more varied types of math activities than did parents with lower income. SES differences have been documented in the frequency of engaging in specific activities that involve math, such as playing with board games or puzzles [32,33]. However, Tudge and Doucet [34] found no differences in

the frequency of math activities by SES, and several studies have even observed negative associations between SES indicators and the HNE, such that these practices are more frequent among families of low SES compared to families of high SES [19,35]. These mixed findings may stem from differences in the underlying dimensions of HNE measured across studies. As described above, past work has primarily distinguished between formal and informal numeracy activities, and some evidence indicates that parents with high SES are more likely than parents with low SES to prioritize informal learning opportunities over more formal ones [31,36]. Thus, although parents with higher SES appear to engage in more numeracy activities in the home, particularly in informal contexts, these findings are not consistently replicated and may indicate that SES relates to different aspects of the HNE in divergent ways.

In both of these bodies of literature, insufficient attention has been paid to the way that the unique aspects of SES, most commonly income and parental education, function to predict parental enrichment. Although related, income and educational attainment likely relate to parental behaviors through distinct mechanisms. Income likely constrains parents' resources and their ability to invest in their children, including purchasing materials for children, providing children with enriching experiences, and spending time with children (i.e., to offset opportunity costs of spending time away from the labor market [37,38]). Consistent with this framing, the parental investment theory explains that income differences in parenting and child outcomes emerge when low-income families experience scarce resources and are less able to invest time and money in their children [39–41]. However, resource strain may also take a psychological toll on parents, such that low income relates to higher levels of stress, resulting in strained relationships, family dysfunction, and emotional distress [40,41].

In contrast, it is less clear how educational attainment may influence the home learning environment. Past work suggests that attaining higher levels of education may serve as an opportunity to acquire cultural capital that shapes parents' values and approaches to parenting [42,43], but how this process unfolds remains unknown. Additionally, educational attainment may confer knowledge and skills to parents, such as increased access to written materials or professionals for parenting advice or more accurate knowledge of child development, which could support more developmentally appropriate behaviors [26,44,45]. However, as past research typically has not distinguished between income and education when examining SES differences in the HLE and HNE, more work is needed to disentangle how each operates to predict parents' home literacy and numeracy enrichment practices.

## 4. Parental Beliefs and Home Enrichment

In addition to parental socioeconomic resources such as income and educational attainment, growing evidence indicates that psychological factors, including parents' beliefs and cognitions, may also shape parental behaviors. Both the Home Literacy and Home Numeracy models [16,18] proposed that parental beliefs and attitudes, including domain-specific attitudes towards math or reading as well as domain-general expectations for children's learning, should relate to enrichment activities in these domains. A large body of research has examined how parents' expectations for their children's academic success relate to enrichment and generally finds strong links between expectations and behaviors [46]. However, less is known about how parents' beliefs about academics and specific academic domains might relate to home enrichment practices. In this study, we examined parents' beliefs about the importance of specific academic subjects (or skill importance beliefs) as well as their beliefs about whether parents or teachers are more important for teaching these skills (or parental responsibility beliefs). These types of beliefs have been relatively understudied in past work, but evidence indicates that both may relate to home enrichment practices. Past work examining skill importance beliefs finds that parents who believe academic achievement is important engaged in enrichment activities more frequently than parents who report lower ratings of the importance of these early academic skills [13,15,47,48]. Fewer studies have examined the links between parental

responsibility beliefs and enrichment, as most of the research in this area has examined differences in beliefs based on SES. The work by Lareau [49] suggests that parents of low SES perceive themselves as less important for teaching their children academic skills than teachers. However, other studies demonstrate that parents of low SES nonetheless value being involved in their child's learning at high rates and these beliefs are positively related to practices to support children's academic achievement [50], indicating that skill importance beliefs and parental responsibility beliefs may both relate to parental behaviors.

The question of how domain-specific skill importance beliefs and parental responsibility beliefs relate to the HLE and HNE in particular still remains largely unanswered. Most work examining beliefs about literacy combines these two dimensions of parental beliefs into a single scale, capturing beliefs about the usefulness of reading, parental responsibility for teaching language and literacy skills, and the importance of these activities; these studies typically find that positive beliefs relate to the frequency of literacy activities at home [51–53]. For example, Bingham [54] measured parents' beliefs about children's literacy development, which included beliefs about what parents should do at home to help their children learn to read as well as what children learn from reading, and found that this composite measure was positively related to parents' reports of the HLE. However, few studies have disaggregated these beliefs into separate measures, and so it is unknown how skill importance beliefs and parental responsibility beliefs may independently relate to parents' home enrichment practices.

Additionally, the evidence regarding relations between beliefs about math and the HNE is quite mixed. Although some studies document positive correlations between how important parents thought it was for children to do math activities at home and the reported frequency of these activities in early elementary school [55], others have found that these associations did not extend to children's math achievement [56], rendering interpretations of these results difficult. In addition, Musun-Miller and Blevins-Knabe [56] found that math activities were more strongly related to beliefs about other academic skills than they were to beliefs about math, suggesting that beliefs about the importance of math may not be unique predictors of math skills. Past work also calls into question domain-specific pathways between parental responsibility beliefs and the HNE. In several studies, parents' beliefs about the relative influence of the home environment compared to the school environment for teaching math did not predict the HNE [31,56]. Together, these complex patterns of findings underscore the need for more work examining domain-specific links between parental beliefs and unique dimensions of the HLE and HNE.

## 5. The Current Study

The extant literature indicates that the HLE and HNE are important aspects of the home environment for young children's academic skill development and that individual variability in parents' provision of reading and numeracy activities is likely linked to both socioeconomic as well as psychological factors. However, questions regarding the nature of these processes remain, such as the underlying dimensions of literacy and numeracy enrichment at home as well as how more nuanced measures of SES and parental beliefs relate to these distinct dimensions. To address these limitations of the existing research base, this study examined two research questions. First, what are the distinct factors of home literacy and numeracy practices reported by parents of young children? Second, how do socioeconomic resources such as education and income as well as psychological characteristics such as parental beliefs about the importance of academic skills and their role in teaching their children predict these aspects of the home environment? This study utilized data from a heterogeneous, nationwide sample of US parents of three- to six-year-old children to answer these questions.

## 6. Methods

### 6.1. Participants

Participants in this study included 259 parents of three- to six-year-old children recruited through Amazon's Mechanical Turk. Originally, 851 participants completed the screener in Qualtrics. However, 406 of these individuals did not pass at least one of the quality checks included in the survey (e.g., "Choose any response other than agree") and so were excluded from these analyses. Additionally, 158 participants lived outside of the U.S. and an additional 24 parents reported on a child outside of the age range given. Listwise-deletion was used to handle item-level missing data (child gender or ratings of how responsible they felt for teaching their child math, $n = 4$), resulting in a final sample of 259 parents. Descriptive statistics for this sample are shown in Table 1. Notably, this sample appears to have slightly higher levels of education than the larger U.S. population of parents of kindergarteners. Only 13% of parents in this sample had a high school diploma or less compared to 29% in nationally representative samples. However, the percentages of parents with a Bachelor's degree or more were comparable (39% compared to 38%) [57].

**Table 1.** Descriptive statistics for key study variables, $n = 259$.

| Variable | M (SD)/% |
|:---:|:---:|
| Parent Education | |
| High School or Less | 13% |
| Some College | 48% |
| Bachelors | 29% |
| Graduate | 10% |
| Income (in 10,000's) | 6.36 (3.73) |
| Parental Responsibility (1–5 scale) | |
| Math Skills | 1.53 (1.14) |
| Communication Skills | 2.51 (1.06) |
| Literacy Skills | 1.98 (1.10) |
| Beliefs about Skill Importance | |
| Math | 2.99 (0.70) |
| Literacy | 3.02 (0.69) |
| Race/Ethnicity | |
| White | 57% |
| Black | 14% |
| Latino | 13% |
| Other | 16% |
| Child is Male | 52% |
| Parent is Male | 40% |
| Child Age (in years) | 4.79 (1.06) |

### 6.2. Procedure

All data were collected through Amazon's Mechanical Turk (MTurk), an online crowd-sourcing platform for tasks that require human intelligence. In general, recruiters post tasks on MTurk, which workers, i.e., any individual across the world with access to the internet who registers as a worker on MTurk, can choose to complete; if the work is satisfactory, then the recruiter can grant approval and payment to the worker. MTurk has grown as a method for psychological research over the past several years with studies demonstrating that MTurk generates reliable data and socioeconomically diverse samples [58]. Although online surveys have certain limitations in that the sample is most likely not representative of the broader population, we believe MTurk's ability to transcend regional and geographic barriers by including participants outside the researchers' physical location is invaluable.

The survey was designed on Qualtrics and then attached as a link to the MTurk survey. Parents who qualified for the MTurk survey were allowed to proceed to the Qualtrics link. As the first step of the survey, parents provided informed consent for participation in this study, and this project was approved by the Institutional Review Board of the University

of Pittsburgh, project number PRO16110252. All data were collected between November of 2017 and March of 2018.

*6.3. Measures*

6.3.1. Home Literacy and Numeracy Enrichment

Parents were asked to report how often they had engaged in a variety of different activities at home, including both literacy and numeracy activities. For each item, parents were asked to report how frequently they had engaged in the activity with their child on a scale from 1 (never) to 5 (almost daily). Specifically, 24 items addressing parental support for numeracy and 8 items addressing support for reading were presented to parents (see Tables 2 and 3). A wide range of items drawn from a number of past surveys [8,19,22,59] was used in this study, particularly for numeracy enrichment, in an attempt to capture the full range of types of activities to support learning that may occur in the homes of young children.

**Table 2.** Item factor loadings for home literacy activities.

| Item | Passive | Active | Communality |
|---|---|---|---|
| Identifying names of written alphabet letters | 0.91 | | 0.77 |
| Identifying sounds of alphabet letters | 0.81 | | 0.69 |
| Teaching your child new words at home | 0.38 | | 0.46 |
| Reading books to your child | 0.35 | | 0.19 |
| Printing letters | | 0.49 | 0.16 |
| Having your child read books | | 0.52 | 0.24 |
| Writing letters with your child | | 0.66 | 0.48 |
| Teaching the correct way to use writing utensils | | 0.48 | 0.33 |

Note. Factor loadings are based on the rotated solution, with values below 0.30 not shown in the table.

**Table 3.** Item factor loadings for home numeracy activities.

| Item | Applications | Basic | Written | Communality |
|---|---|---|---|---|
| Learning simple sums | 0.47 | | | 0.46 |
| Talking about money when shopping | 0.63 | | | 0.48 |
| Measuring ingredients when cooking | 0.74 | | | 0.49 |
| Being timed | 0.50 | | | 0.27 |
| Playing with a calculator | 0.50 | | 0.31 | 0.44 |
| Talking about time with clocks and calendars | 0.51 | | | 0.44 |
| Helping your child weigh, measure, and compare quantities | 0.81 | | | 0.64 |
| Playing games that involve counting, adding, or subtraction | 0.32 | | | 0.36 |
| Measuring lengths/widths | 0.73 | | | 0.54 |
| Using a computer to learn math | 0.44 | | | 0.35 |
| Identifying names of written numbers | | 0.45 | | 0.35 |
| Playing with number fridge magnets | | 0.32 | | 0.19 |
| Counting objects | | 0.90 | | 0.69 |
| Sorting things by size, color, or shape | | 0.71 | | 0.59 |
| Counting down | | 0.41 | | 0.27 |
| Making collections | | 0.47 | | 0.39 |
| Asking about quantities/how many | | 0.65 | | 0.47 |
| Singing counting songs | | 0.49 | | 0.24 |
| Reading number storybooks | | 0.33 | 0.46 | 0.37 |
| Using number or arithmetic flashcards | | | 0.59 | 0.42 |
| Printing numbers | | | 0.37 | 0.38 |
| Connect the dot activities | | | 0.73 | 0.56 |
| Using number activity books | | | 0.67 | 0.57 |
| Playing card games | | | 0.44 | 0.40 |

Note. Factor loadings are based on the rotated solution, with values below 0.30 not shown in the table.

### 6.3.2. Parental Responsibility for Learning

Parents were also asked to report the relative importance of the home and school contexts for children's learning during elementary school. Specifically, parents were asked to rate nine different domains of skills (e.g., character/moral development, and physical fitness) on a scale of 1, indicating that the parents' influence was much more important than the school's influence, to 5, indicating that the school's influence was much more important than the parents' influence. These items were drawn from the Skill Responsibility Survey [60]. For the purposes of this study, we examined parents' reports of the relative importance of the home and school context for each of the three cognitive domains of development: verbal communication, literacy, and math. For each, parents' responses were reverse coded, such that higher scores reflected more perceived parental responsibility for that domain and lower scores reflected more perceived school responsibility for that domain. Each domain included a single item, and so internal consistency could not be calculated.

### 6.3.3. Parental Beliefs about Skill Importance

Parents also reported their expectations for children's literacy and math skills at school entry through a series of ten items describing specific literacy benchmarks (e.g., reciting the alphabet, printing first name) and ten items describing math benchmarks (e.g., counting to 100, writing numbers 1 through 5) [8]. For each item, parents were asked to rate the importance of this skill for children entering kindergarten on a scale from 1 (not important) to 5 (very important). This scale was adapted to include a broader range of skills but retained good internal reliability ($\alpha = 0.91$ and $\alpha = 0.86$ for reading and math skills, respectively).

### 6.3.4. Socioeconomic Status

Parents were also asked to report their household income and their level of educational attainment. Total income from all household members over the past year was reported by selecting the range that contained their income rather than reporting an exact dollar amount. Ranges increased by USD 10,000 increments from USD 10,000 or less to USD 200,001 or more. Parents also reported their level of education by selecting the highest degree or certificate that they held. Responses were recoded into four categories: high school (i.e., none, high school diploma or GED), some college (i.e., nursing certificate, some college but no degree, Associate's degree), Bachelor's degree (i.e., Bachelor's degree, some graduate work), or Graduate degree (i.e., Master's degree, M.D., Ph.D., Law, Dental, or other advanced degree).

### 6.3.5. Control Variables

In addition to the key variables of interest described above, parents were asked to report their child's gender and date of birth, which was used to calculate their children's exact age. Parents also reported their own gender and their race/ethnicity; responses were coded into four categories: White, Black, Hispanic/Latino, and other, which included parents who reported more than one race, Asian or Pacific Islander, American Indian or Alaskan Native, or other.

### *6.4. Analytic Plan*

A series of exploratory factor analyses were conducted in Stata 14 using the iterated principal factor with promax oblique factor rotations and Horst normalization [61]. Factor analyses were estimated separately for home literacy and home numeracy items. Kaiser–Meyer–Olkin (KMO) Measures of Sampling Adequacy and Bartlett tests of sphericity were also calculated for each set of items to ensure that factor analyses were appropriate [62,63]. Selection of the appropriate number of factors was based on interpretability and clarity of the factors as well as factor eigenvalues. Composite variables were then calculated for each literacy and numeracy factor, with cross-loading items (i.e., items with rotated factor

loadings greater than 0.30 on two or more factors) excluded from composites. A series of OLS regression models were then estimated predicting each factor first from SES variables and the set of control variables, with parental beliefs variables included in a second step of the regressions.

## 7. Results

### 7.1. Dimensions of Home Enrichment Practices

To address our first research question, what distinct factors of home literacy and numeracy practices are reported by parents of young children, we estimated a series of factor analyses. A two-factor solution was selected as the most appropriate method of accounting for variability in parents' home literacy practices. Items and factor loadings are shown in Table 2. The first factor, labeled passive literacy activities, included items such as identifying the names of and sounds associated with letters of the alphabet. In contrast, the second factor included items such as having the child read or write and, as such, was labeled active literacy activities. As shown in Table 4, these factors were moderately correlated, suggesting that parents engaged in similar levels of active and passive activities but that these factors, nonetheless, reflected unique aspects of the home literacy environment. The KMO Measure of Sampling Adequacy for these items was 0.79 and the Bartlett test of sphericity was significant, $\chi^2(28) = 571.21$, $p < 0.001$, suggesting that factor analyses were appropriate with these data.

**Table 4.** Descriptive statistics of home literacy and home numeracy subdimensions.

|  | Passive Literacy | Active Literacy | Numeracy Applications | Basic Numeracy | Written Numeracy |
|---|---|---|---|---|---|
| M | 3.99 | 3.25 | 2.67 | 3.45 | 2.87 |
| SD | 0.85 | 0.97 | 0.90 | 0.80 | 0.94 |
| Skewness | −0.60 | −0.31 | 0.17 | −0.21 | −0.06 |
| Kurtosis | 2.35 | 2.44 | 2.39 | 2.43 | 2.31 |
| N of items | 4 | 4 | 9 | 8 | 5 |
| Alpha | 0.73 | 0.69 | 0.86 | 0.81 | 0.79 |
| **Correlations** |  |  |  |  |  |
| Passive Literacy | 1.00 |  |  |  |  |
| Active Literacy | 0.51 | 1.00 |  |  |  |
| Numeracy Applications | 0.34 | 0.58 | 1.00 |  |  |
| Basic Numeracy | 0.63 | 0.47 | 0.48 | 1.00 |  |
| Written Numeracy | 0.41 | 0.68 | 0.65 | 0.52 | 1.00 |

Note. All correlations are significant at $p < 0.001$.

A three-factor solution was selected as most appropriate for the home numeracy items (see Table 3 for items and factor loadings). The first factor, labeled numeracy applications, included day-to-day activities that involved math, including talking about money when shopping and measuring. The second was labeled basic numeracy activities and included items that targeted basic counting and number identification skills, such as singing counting songs and playing with number fridge magnets. Finally, the third factor was labeled written numeracy activities and included activities such as number activity books as well as card games. Two items had more than one factor loading above 0.30: "playing with a calculator" loaded on both numeracy applications and written numeracy activities, and "reading number storybooks" loaded on both basic numeracy activities and written numeracy activities. As noted in the analytical plan, these items were not included in composites. Correlations between the three factors ranged from 0.48 to 0.65, as shown in Table 4. The KMO Measure of Sampling Adequacy (0.90) and Bartlett test of sphericity ($\chi^2(278) = 2507.44$, $p < 0.001$) for these items also indicated that factor analyses were appropriate.

## 7.2. Predictors of Home Enrichment Practices

To address our second research question, how socioeconomic resources and psychological characteristics predict aspects of the home environment, we estimated a series of regression models predicting each domain of home literacy and home numeracy.

### 7.2.1. Home Literacy

Results from regression models predicting each dimension of the home literacy environment are shown in Table 5. Passive literacy activities were unrelated to both income and parental education, but beliefs about the importance of literacy skills were positively and significantly predictive of these activities. Specifically, a 1 standard deviation (SD) increase in beliefs about the importance of literacy related to a 0.23 SD increase in passive literacy activities.

**Table 5.** Regression results predicting each dimension of home literacy practices from SES variables, parental beliefs, and covariates.

| | Passive | | Active | |
|---|---|---|---|---|
| | B (SE) | B (SE) | B (SE) | B (SE) |
| Parent Education | | | | |
| Some College | 0.04 (0.16) | 0.09 (0.16) | −0.45 * (0.18) | −0.32 [†] (0.18) |
| Bachelors | 0.02 (0.18) | 0.09 (0.17) | −0.37 [†] (0.20) | −0.24 (0.19) |
| Graduate | 0.07 (0.23) | 0.10 (0.23) | −0.54 * (0.26) | −0.49 * (0.25) |
| Income | 0.02 (0.01) | 0.02 (0.01) | 0.04 * (0.02) | 0.03 [†] (0.02) |
| Parental Responsibility | | | | |
| Math Skills | | −0.05 (0.06) | | 0.05 (0.06) |
| Communication Skills | | 0.04 (0.05) | | −0.12 * (0.06) |
| Literacy Skills | | 0.09 (0.06) | | 0.15 * (0.07) |
| Beliefs about Skill Importance | | | | |
| Math | | 0.002 (0.12) | | 0.17 (0.13) |
| Literacy | | 0.29 ** (0.12) | | 0.23 [†] (0.13) |
| Race/Ethnicity | | | | |
| Black | −0.35 * (0.15) | −0.38 * (0.15) | −0.03 (0.17) | −0.06 (0.16) |
| Latino | −0.22 (0.15) | −0.26 [†] (0.15) | −0.16 (0.18) | −0.19 (0.17) |
| Other | −0.04 (0.14) | −0.06 (0.14) | 0.19 (0.17) | 0.17 (0.16) |
| Child is Male | 0.04 (0.10) | 0.05 (0.10) | −0.08 (0.12) | −0.10 (0.11) |
| Parent is Male | −0.57 *** (0.11) | −0.52 *** (0.11) | −0.40 ** (0.12) | −0.39 ** (0.12) |
| Child Age | 0.01 (0.05) | −0.001 (0.05) | 0.27 *** (0.06) | 0.25 *** (0.05) |
| Intercept | 4.07 *** (0.27) | 3.00 *** (0.39) | 2.29 *** (0.31) | 1.08 ** (0.43) |

[†] $p < 0.10$, * $p < 0.05$, ** $p < 0.01$, *** $p < 0.001$.

In contrast, active literacy activities were related to both income and education, albeit in different directions. Compared to parents who completed high school or less, parents who attended some college, attained a Bachelor's degree, or completed a graduate program reported 0.45, 0.38, and 0.56 SDs less active literacy activities at home with their young children, although this contrast was only marginally significant for parents with a Bachelor's degree. Additionally, a 1 SD increase in income was associated with a 0.14 SD increase in active literacy activities. In terms of parental beliefs, parents' reports of how responsible they felt for developing their children's communication and literacy skills were also related to these active practices, as a 1 SD increase in responsibility for literacy related to a 0.17 SD increase in active literacy practices, whereas a 1 SD increase in responsibility for communication predicted a 0.13 SD decrease in these practices. Finally, a non-significant trend emerged for beliefs about the importance of literacy skills, as a 1 SD increase in these ratings related to a 0.16 SD increase in active literacy activities.

In addition to these hypothesized predictors, several associations emerged between covariates and these measures of home literacy practices. Fathers reported significantly fewer passive and active literacy activities (0.61 SD and 0.40 SD fewer than mothers, respectively, in the fully controlled models). Additionally, parents of older children reported

significantly more active literacy activities, as a 1 SD increase in age was associated with a 0.27 SD increase in active literacy activities.

### 7.2.2. Home Numeracy

Results from regression models predicting dimensions of home numeracy activities are shown in Table 6. Parental engagement with math applications was unrelated to socioeconomic status, including both income and education. Instead, parental responsibility for teaching math was associated with higher levels of these activities, as a 1 SD increase in parental reports of responsibility related to a 0.19 SD increase in math applications. Responsibility for helping develop children's communication skills and beliefs about the importance of math also marginally related to application activities ($\beta = -0.13$ and $\beta = 0.16$, respectively), but these associations failed to reach conventional significance levels.

**Table 6.** Regression results predicting each dimension of home numeracy practices from SES variables, parental beliefs, and covariates.

| | Applications | | Basic | | Written | |
|---|---|---|---|---|---|---|
| | B (SE) | B (SE) | B (SE) | B (SE) | B (SE) | B (SE) |
| **Parent Education** | | | | | | |
| Some College | −0.02 (0.17) | 0.09 (0.16) | −0.01 (0.16) | 0.04 (0.15) | −0.35 [†] (0.18) | −0.25 (0.18) |
| Bachelors | 0.01 (0.19) | 0.13 (0.18) | 0.05 (0.17) | 0.13 (0.16) | −0.18 (0.20) | −0.07 (0.19) |
| Graduate | 0.04 (0.25) | 0.07 (0.23) | −0.29 (0.22) | −0.28 (0.21) | −0.57 * (0.26) | −0.56 * (0.25) |
| Income | 0.01 (0.02) | 0.003 (0.01) | 0.02 (0.01) | 0.02 (0.01) | 0.02 (0.02) | 0.02 (0.02) |
| **Parental Responsibility** | | | | | | |
| Math Skills | | 0.15 * (0.06) | | 0.06 (0.05) | | 0.14 * (0.06) |
| Communication Skills | | −0.11 [†] (0.06) | | 0.06 (0.05) | | −0.07 (0.06) |
| Literacy Skills | | 0.11 (0.06) | | 0.03 (0.06) | | 0.04 (0.07) |
| **Importance Beliefs** | | | | | | |
| Math | | 0.20 [†] (0.12) | | 0.07 (0.11) | | 0.17 (0.13) |
| Literacy | | 0.18 (0.12) | | 0.25 * (0.12) | | 0.20 (0.13) |
| **Race/Ethnicity** | | | | | | |
| Black | −0.03 (0.17) | −0.04 (0.15) | −0.11 (0.15) | −0.14 (0.14) | −0.01 (0.17) | −0.04 (0.17) |
| Latino | 0.15 (0.17) | 0.15 (0.16) | −0.06 (0.15) | −0.07 (0.15) | 0.17 (0.17) | 0.17 (0.17) |
| Other | 0.28 [†] (0.16) | 0.24 (0.15) | 0.01 (0.14) | −0.02 (0.14) | 0.30 [†] (0.17) | 0.28 [†] (0.16) |
| Child is Male | 0.10 (0.11) | 0.08 (0.10) | −0.01 (0.10) | 0.03 (0.10) | 0.09 (0.12) | 0.08 (0.11) |
| Parent is Male | −0.05 (0.11) | −0.04 (0.11) | −0.41 *** (0.10) | −0.36 *** (0.10) | −0.13 (0.12) | −0.12 (0.12) |
| Child Age | 0.24 *** (0.05) | 0.23 *** (0.05) | −0.06 (0.05) | −0.07 (0.05) | 0.21 *** (0.06) | 0.20 *** (0.05) |
| Intercept | 1.36 *** (0.30) | 0.07 (0.40) | 3.83 *** (0.27) | 2.56 *** (0.37) | 1.93 *** (0.31) | 0.75 [†] (0.43) |

[†] $p < 0.10$, * $p < 0.05$, *** $p < 0.001$.

Similarly, income and educational attainment were unrelated to parental reports of basic numeracy activities. Of the parental beliefs, only parents' reports of how important they considered early literacy skills to be were significantly related to basic numeracy activities. Specifically, a 1 SD increase in beliefs about the importance of early literacy skills was associated with a 0.21 SD increase in basic numeracy activities.

Finally, parental education was negatively related to reports of written numeracy activities. Parents with a graduate degree reported engaging in 0.61 SD lower levels of written numeracy activities than parents without any additional education after high school. Similar to the results from the applications factor, parents who rated their responsibility for teaching their children math higher tended to report more written math activities at home, as a 1 SD increase in beliefs about parental responsibility predicted a 0.17 SD increase in written numeracy activities.

As in the models predicting literacy activities, several interesting associations were seen between parents' numeracy activities and control variables. Fathers reported significantly fewer basic math activities than mothers (0.45 SDs lower in fully controlled models), but no parental gender differences were seen in reports of numeracy applications and written numeracy activities. Instead, child age was significantly related to both numeracy

applications and written numeracy activities, as a 1 SD increase in age was associated with a 0.27 SD increase in numeracy applications and a 0.22 SD increase in written numeracy activities.

## 8. Discussion

In this study, we examined the dimensionality of the home literacy and numeracy environment and sought to explain individual variability in these aspects of home enrichment. Specifically, we addressed two research questions. First, what factors of home literacy and numeracy practices are reported by parents of young children? Using data collected from parents of young children through Amazon's MTurk, we identified two dimensions of the HLE—active and passive literacy activities—as well as three dimensions of the HNE—numeracy applications, basic numeracy, and written numeracy. Second, how do socioeconomic and psychological factors relate to home literacy and numeracy practices? SES variables were related to active literacy activities, although in complementary directions, whereas few significant SES differences were seen in parents' numeracy activities. Additionally, several domain-specific associations between parental beliefs and enrichment activities were seen. Below, we discuss each of these findings in detail as well as directions for future work, such as associations among predictors of the home learning environment and alternative measures of how parents support children's academic skills.

### 8.1. Dimensions of the Home Literacy and Numeracy Environment

Two distinct dimensions of literacy activities were identified in our analyses of the home literacy items: active and passive literacy activities. Past research has utilized similar names to describe dimensions of literacy activities [64], but in this past work, "active" literacy was defined as activities that children engaged in, whereas "passive" literacy included those activities that children simply observed (e.g., watching a parent read). By this definition, all literacy activities included in this scale would be considered active. Instead, the dimensions identified in the present study were differentiated by the extent to which children themselves were independently contributing to the activity such as by writing.

This conceptualization of literacy practices at home is quite dissimilar to the commonly supported framework of formal and informal literacy activities. Despite some potential overlap in these categorizations, formal literacy activities such as teaching children to write letters could include both passive (e.g., identifying the names of letters) and active (e.g., having children write letters) components. Similarly, informal activities such as shared book reading could be passive in some instances, such as a parent reading to a child, or active, in the case of a child reading to the parent. Many literacy activities may include fluctuation between passive and active components throughout the natural progression of the activity, and so more research replicating these dimensions and exploring how active and passive literacy activities naturally arise is critical. Furthermore, it is important to examine the ways that parents' engagement in each of these types of activities may depend on the child's own skill. Many parents may transition from reading with their child to having their child read to them as the latter becomes more developmentally appropriate. In a similar vein, past studies have distinguished between literacy activities based on whether the activities target basic or advanced literacy skills (e.g., recognizing letters compared to reading words) [65], with parents presumably adjusting their activities based on the needs of their child. Considering these measures of the HLE as a snapshot of parents' current practices, based not only on the SES and beliefs variables included as predictors in these models but also the needs of the child at this particular time, is important for capturing the complex nature of the HLE.

Similarly, the dimensions of the HNE identified in this study do not replicate the frequently replicated distinction between formal and informal math activities described in the Home Numeracy Model [18], but many of the factors identified here have, nonetheless, been described in past work. LeFevre and her colleagues [8] describe a dimension of

numeracy applications that included similar types of activities as included here, and other researchers have analyzed measures of basic numeracy activities [22,23]. However, to our knowledge, past work has not explored numeracy activities that explicitly involve written numbers. The emergence of this factor was particularly interesting given that the activities included both traditionally formal, didactic learning activities (e.g., using flashcards and number activity books) as well as play with written numbers that is typically considered informal (e.g., playing card games). Thus, more research replicating this unique factor and exploring the extent to which interactions with written Arabic numerals differ from numerical activities that occur in the absence of numerals is an important next step.

*8.2. SES and Home Enrichment*

SES was related to several domains of the HLE and HNE. Specifically, parents with higher levels of education typically reported engaging in activities involving written numbers less frequently than their peers. Although this finding is inconsistent with research on the home learning environment more generally [12,14], it is not particularly surprising given evidence suggesting that didactic math activities, such as flashcards and workbooks, are used more frequently by parents with lower levels of education [19]. Specifically, in one study, parents of middle SES reported embedding math into daily activities more so than parents of low SES and also engaged in more made-up math games and incorporated math into routines, whereas parents of low SES were more likely to set time aside to work with children directly to improve math skills [31]. However, it is notable that the written HNE scale included several items that were not didactic or formal in nature, as described above.

SES was also related to parents' reports of active literacy activities, but educational attainment and income operated uniquely and in opposite directions to predict these activities. Parents with higher incomes tended to report active literacy activities more frequently, whereas higher education was associated with lower reported frequency of these activities. Similar to the HNE findings reported above, some evidence does indicate that these parents report higher than average frequencies of specific types of learning activities, particularly those that are didactic in nature [36]. As such, it is possible that these mixed associations reflect multiple unique mechanisms, such that income reduces financial strain, and thus, allows parents to invest time and financial resources to engage in these active literacy activities [37–40], whereas parents with higher levels of education may engage in these specific types of literacy activities less often if they typically provide other types of literacy stimulation at home that is more informal [36]. Regardless of the mechanisms underlying these relations, this complex pattern of findings underscores the importance of differentiating between income and educational attainment in future research examining SES both in sampling as well as data analysis.

Finally, many null associations between SES variables and various components of the HLE and HNE were also observed in these associations. Income and parental education were not related to the applications or basic HNE scales, which is consistent with past studies that have found similar null associations between SES and measures of the HNE [34]. However, passive literacy activities were also not related to either income or parental education, despite decades of research indicating that activities included in this subscale, such as shared book reading, vary systematically depending on parental SES [12,27]. On the one hand, the passive HLE scale included other activities that have been examined less frequently in past work about teaching children letters, and so it is possible that this subscale, which reflected more than just book reading, may not have been associated with SES. This interpretation seems plausible given the relatively low factor loading of the book reading item on this scale. Alternatively, SES may be more related to other qualitative aspects of these literacy activities, such as how long parents participated in the activities or the nature of the interactions, than to the frequency at which these activities occurred, as was measured in this study. Finally, it is possible that recent efforts to encourage literacy activities and increases in public dialogue about the importance of early literacy skills

have narrowed some disparities in parents' literacy activities. More research disentangling these and other possible explanations is necessary for designing studies and interventions addressing SES gaps in children's literacy skills.

How do we reconcile these null or negative associations between SES and dimensions of the HLE and HNE with the abundant literature demonstrating strong, robust links between SES and children's outcomes? These inconsistent associations highlight the need for more research linking these components of the HLE and HNE to children's academic outcomes, particularly for the HNE, where findings have been quite mixed [11,66]. Furthermore, expanding our view beyond activities that occur in the home to include broader aspects of the home environment, such as language input [24,67] or conversations about numbers [21,68,69], is critical to ensure that the full range of opportunities for learning literacy and numeracy skills at home is measured. There may be other activities or types of interactions that parents with high SES engage in more than parents with low SES, and by focusing exclusively on a single domain of the HLE or HNE in isolation, we may overlook qualitative differences in the ways that these behaviors co-occur or cluster among certain types of families.

### 8.3. Parental Beliefs and Home Enrichment

Parents' beliefs about early literacy and math skill development were related to their reports of activities at home in theoretically expected ways, with a few exceptions. Parents who rated literacy skills as more important reported engaging in both passive and active HLE activities more frequently than their peers, although this association failed to reach conventional significance levels for the active HLE scale. In contrast, parents who believed they were more responsible for teaching these skills engaged in more active literacy activities at home. These differences in the patterns of significance across outcomes is not surprising when considering that the active literacy items included here may have involved more direct teaching or intentional activities, and so these activities may be less likely to occur when parents do not feel responsible for initiating them.

Parents' beliefs about the importance of children's math skills were largely unrelated to their numeracy activities at home, whereas parents' beliefs about their own responsibility for teaching their children math were positively related to both the applications and written HNE dimensions. As such, many parents may believe that math skills are important but better left to their children's teacher, consistent with evidence that parents tend to view themselves as less responsible for teaching their children math compared to other academic domains such as literacy [56,60]. These associations demonstrate the importance of measuring many parental beliefs beyond just how important parents believe certain skills to be.

Although most associations between parents' beliefs and their reported activities at home were within a single domain (e.g., beliefs about the importance of literacy skills predicting literacy activities), a significant relation between beliefs about the importance of literacy skills and parents' basic numeracy activities was observed as well. In contrast, parents' beliefs about the importance of math skills were unrelated to basic numeracy activities. There are several possible interpretations of these findings that should be explored in future research. On the one hand, this cross-domain link could demonstrate that parents view the basic numeracy activities included in this study as more closely related to literacy activities, as bivariate correlations between the basic HNE and passive HLE scales, for example, would suggest. In other words, some parents may have rated the more advanced math benchmarks on this survey as less important for school entry but, nonetheless, believed that it was important for their children to develop these more fundamental numeracy skills. Alternatively, parents who believe that literacy skills are very important may emphasize these basic numeracy activities that are easily integrated into other activities as opposed to more structured, numeracy-centered interactions. More work examining the messages that parents receive about supporting literacy and numeracy

skills at home as well as parents' decision making about how to spend their time with their children may shed light on this unexpected association.

### 8.4. Remaining Questions and Conclusions

Several important questions stemming from these results remain unanswered. First, beliefs and SES variables were modeled as independent characteristics in this study, and yet parents' income or level of education may shape parents' beliefs about their children's education. Several studies have shown that low-income parents were more likely than middle-income parents to believe that the preschool classroom was more important than the home environment for children's learning [31,70], possibly due to parents of low SES also showing lower self-reported confidence in their teaching abilities [50,71]. Additionally, Zheng and Libertus [72] found that parents' education was positively related to beliefs about how important it was that their child was exposed to math content in school, whereas parental income was positively related to beliefs about how important it was to parents that their children were exposed to reading and writing activities at home, illustrating divergent findings on SES's influence on parental beliefs. As such, many questions remain regarding how parents' beliefs are formed and the implications for enrichment practices.

We also observed several associations between parent gender and some aspects of literacy and numeracy support in the home. Previous work has shown that fathers may be an especially important source of math learning for children. Fathers' engagement in learning activities with toddlers in a low-SES sample predicted children's math scores in 5th grade [73]. Additionally, although mothers and fathers do not differ in the diversity of their talk with their toddlers, fathers may ask more questions of their children, suggesting some qualitative differences in the nature of their interactions [74]. Fathers also contribute uniquely to the language and cognitive skills of 2- and 3-year-olds above mothers' contributions [75–77]. However, past work also suggests that fathers may engage in learning activities less often than mothers [78]. Although examining differences in how mothers and fathers support children's learning during early childhood was not a central aim of this study, our findings extend this past work indicating specificity in gender differences. Specifically, although gender differences were detected for literacy activities and for basic numeracy activities, no differences between mothers and fathers were seen in written numeracy activities or numeracy applications. More work with a large sample of both mothers and fathers directly addressing the extent to which mothers and fathers differ in their provision of learning opportunities and on which measures of home activities is needed.

More generally, it remains unclear to what extent the measures utilized here capture the full complexity of the home literacy and numeracy environments that children experience. Extant scholarship indicates that parents' beliefs about and children's interest in literacy relate to literacy outcomes independent of parental teaching of literacy in the home, suggesting that creating a home environment to support literacy development means more than simply incorporating literacy activities [65]. Similarly, many recent studies have relied on alternative metrics of opportunities to learn math at home, such as parents' discussion of numbers or other math concepts during interactions with their young children [68,69,79–81]; several studies suggest that parental number talk and activities reflect unique aspects of the home numeracy environment, as correlations between the two measures are often moderate at best [82–84]. Thus, it is critical to consider the ways that the predictor variables included here might relate to broader aspects of the home literacy and numeracy environments beyond the types of activities measured here. Likewise, there may be additional contextual predictors, such as other children in the home or the information and resources that parents receive from their children's school, that may shape parents' practices in meaningful ways or even interact with parents' practices to predict children's skills. As such, more work embedding these practices in the broader family and cultural context is needed.

Several limitations of this study warrant discussion. Although collecting data through Mturk enabled us to collect a large sample of parents from across the U.S., this method also precluded any direct observational measures of parent–child interactions or children's skills. As such, we rely solely on parental reports of all study variables, which may lead to biased reporting that could contribute to the associations documented here. Furthermore, the list of HLE and HNE items may not adequately capture all possible home enrichment activities that families engage in. Most importantly, it remains unknown how each of the dimensions of the HLE and HNE identified in our study would relate to children's academic skills. More generally, a surprisingly large proportion of responses were removed from the sample due to data quality issues such as missing quality checks (as described in the methods) or reporting on a child outside of the dedicated age range. These data quality issues have two major implications. On the one hand, the fact that such a large number of parents failed to correctly respond to simple quality checks such as selecting a designated response suggests that many respondents may have been overly careless when completing the survey. Although these quality checks helped the research team identify many participants who were not closely reading the survey, it is likely that other participants who provided unreliable data may have been missed (e.g., if a participant read the quality checks closely, by chance, but skimmed other questions on the survey). On the other hand, it is possible that some parents, such as those with lower reading comprehension, may have been more likely to respond to these quality check questions incorrectly, and thus, may be underrepresented in this sample. More research examining these tradeoffs in data quality and generalizability is needed to inform future work utilizing Mturk and other large-scale online survey techniques. Despite these limitations, this work adds a new perspective for considering the ways that parents support their young children's literacy and numeracy skills and demonstrates that parents' beliefs about these domains may be consistent with their practices to support each, but socioeconomic differences in the HLE and HNE may be more complex than previously considered.

**Author Contributions:** Conceptualization, L.E., P.Z. and M.L.; Data curation, L.E. and P.Z.; Formal analysis, L.E.; Funding acquisition, P.Z.; Supervision, M.L.; Writing—original draft, L.E.; Writing—review and editing, P.Z. and M.L. All authors have read and agreed to the published version of the manuscript.

**Funding:** This research was funded by the University of Pittsburgh Honors College.

**Institutional Review Board Statement:** The study was conducted according to the guidelines of the Declaration of Helsinki and approved by the Institutional Review Board of the University of Pittsburgh (PRO16110252).

**Informed Consent Statement:** Informed consent was obtained from all subjects involved in the study.

**Data Availability Statement:** The data presented in this study are available on request from the corresponding author.

**Conflicts of Interest:** The authors declare no conflict of interest.

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
