# Peer review of "Individual Differences in Parental Support for Numeracy and Literacy in Early Childhood"

_education, doi:10.3390/educsci11090541_

Round 1

Reviewer 1 Report

The article presents an extensive references overview to document the research hypotheses. The statistical approach presents different correlations discussed in detail, mainly from the statistical perspective.

Certain improvements may be applied for facilitating the reader understanding.

Concerning the descriptive statistics of the study sample, should the parent education (10%/48%/29%/10%) reflect the heterogenous distribution of the targeted population ? Maybe in certain families more than one child are present and, in these conditions, what influence may be seen or at least comment this issue ? In the discussed child age range, school educational system and socio environment have a real influence, too. Maybe a discussion about how these factors balance with parental support should add value.

Concerning the results discussion, extensive statistical presentation appears unbalanced versus content hypotheses validation and interpretation of results. Comparison to other authors may be relevant.

If possible, graphical presentation of results may improve the article overall quality for facilitating the readability.

In the conclusion section, re-iterate the starting hypotheses (questions) and summarize accordingly the obtained results and finally add remaining questions and further research persectives.

Author Response

Concerning the descriptive statistics of the study sample, should the parent education (10%/48%/29%/10%) reflect the heterogenous distribution of the targeted population ?

Details regarding how this sample compares to the broader US population of parents of young children have been added to the participants sections.

Maybe in certain families more than one child are present and, in these conditions, what influence may be seen or at least comment this issue ? In the discussed child age range, school educational system and socio environment have a real influence, too. Maybe a discussion about how these factors balance with parental support should add value.

We thank the Reviewer for this helpful suggestion. A brief discussion of these broader contextual factors is now included in the discussion section on page 31.

Concerning the results discussion, extensive statistical presentation appears unbalanced versus content hypotheses validation and interpretation of results. Comparison to other authors may be relevant.

We have included interpretation of the results in the Discussion section primarily, but we have added description of the research questions in the Results section as well.

If possible, graphical presentation of results may improve the article overall quality for facilitating the readability.

We have included results in the text and Tables for better transparency of our findings (i.e., so that exact estimates can be see and used to compare effect sizes across studies), but we welcome suggestions for additional information to be shown in a figure if the Reviewer has specific data in mind that would further improve the readability of the manuscript.

In the conclusion section, re-iterate the starting hypotheses (questions) and summarize accordingly the obtained results and finally add remaining questions and further research persectives.

We have added more explicit reference to the research questions and results related to each in the discussion as well as the summary of future work, particularly on page 23.

Reviewer 2 Report

Dear authors,

Congratulations on the article. It is an interesting topic analyzed in a professional manner.

Gabriela Neagu

Author Response

We thank the Reviewer for their positive evaluation of our work.

Reviewer 3 Report

Dear Author(s),

I found your study well designed, and the analyses were sound.

Please find below some suggestions.

Participants, measures, procedure and analytical plan are subsections of the Method section and I suggest numbering them accordingly.

Please discuss the reliability of the scales included in your questionnaire.

Please discuss the assumption testing for FA. In the results, describe how you went about testing the assumptions for FA. Details regarding Measures of Sampling Adequacy should be reported.

Lines 280-281: A series of exploratory factor analyses were conducted in Stata 14 using the iterated 280 principal factor with promax oblique factor rotations and Horst normalization [60]. Please explain why PCA was used.

Were there any items removed from the factor analyses?

The tables reporting factor loadings should also report the communality for each variable (in the final column).

Following the presentation of the factor analysis results, reliability analyses should be provided. Reporting of reliability analyses can be combined with a descriptives table which includes names of the factors, the number of items in each factor, descriptive statistics for the composite scores (e.g. mean, SD, Skewness and Kurtosis), and the Cronbach’s alpha (α).

The key tables should be included in the respective sections where they are discussed.

Author Response

Participants, measures, procedure and analytical plan are subsections of the Method section and I suggest numbering them accordingly.

We have added numbers to the subsection headings in the Methods in line with this suggestion.

Please discuss the reliability of the scales included in your questionnaire.

Reliability of the home learning composites is now included in Table 4, and internal reliability statistics for parent beliefs about skill importance are provided in the text. Other questionnaire measures such as the parental responsibility beliefs were measured with a single item, and so internal reliability could not be calculated (and past work has not addressed other indices of reliability such as test-retest or interrater reliability for these constructs). This concern is now briefly noted when describing this scale.

Please discuss the assumption testing for FA. In the results, describe how you went about testing the assumptions for FA. Details regarding Measures of Sampling Adequacy should be reported.

Kaiser-Meyer-Olkin Measures of Sampling Adequacy and Barlett tests of sphericity are now reported in the text for both sets of items that were included in our factor analyses.

Lines 280-281: A series of exploratory factor analyses were conducted in Stata 14 using the iterated 280 principal factor with promax oblique factor rotations and Horst normalization [60]. Please explain why PCA was used.

Principal component analysis was not used in this study, as factors were estimated with the iterated principal factor approach. PCA would not be appropriate in these analyses given that we did not expect the factor structure to fully explain item-level variance (communalities of 1.0).

Were there any items removed from the factor analyses?

No additional items describing home activities were included in the survey or removed from the factor analyses. However, as noted in the text on p. 14, the two numeracy items with crossloadings were not included in home numeracy composites.

The tables reporting factor loadings should also report the communality for each variable (in the final column).

Communalities have been added to Tables 2 and 3.

Following the presentation of the factor analysis results, reliability analyses should be provided. Reporting of reliability analyses can be combined with a descriptives table which includes names of the factors, the number of items in each factor, descriptive statistics for the composite scores (e.g. mean, SD, Skewness and Kurtosis), and the Cronbach’s alpha (α).

Table 4 has been modified to include descriptive information (moved from Table 1) as well as skewness/kurtosis and scale reliability.

The key tables should be included in the respective sections where they are discussed.

The tables have been moved from the end of the document to the main text.